# The meaning of caring for patients with cancer among traditional medicine practitioners in Uganda: A grounded theory approach

**John Baptist Asiimwe**[1,2]*, **Prakash B. Nagendrappa**[3], **Esther C. Atukunda**[1], **Grace Nambozi**[4], **Casim Umba Tolo**[1], **Patrick E. Ogwang**[1], **Maud M. Kamatenesi**[5]

**1** Department of Pharmacy, Mbarara University of Science and Technology, Mbarara, Uganda, **2** Aga Khan University, Uganda Campus, Kampala, Uganda, **3** The University of Trans-disciplinary Health Sciences and Technology, Bangalore, India, **4** Department of Nursing, Mbarara University of Science and Technology, Mbarara, Uganda, **5** Kampala International University, Kampala, Uganda

* john.asiimwe@aku.edu

**Data Availability Statement:** All data are in the manuscript and/or supporting information files.

## Abstract

Traditional medicine practitioners (TMPs) are a critical part of healthcare systems in many sub-Saharan African countries and play vital roles in caring for patients with cancer. Despite some progress in describing TMPs' caring experiences in abstract terms, literature about practice models in Africa remains limited. This study aimed to develop a substantive theory to clarify the care provided by TMPs to patients with cancer in Uganda. This study adhered to the principal features of the modified Straussian grounded theory design. Participants were 18 TMPs caring for patients with cancer from 10 districts in Uganda, selected by purposive and theoretical sampling methods. Researcher-administered in-depth interviews were conducted, along with three focus group discussions. Data were analyzed using constant comparative analysis. The core category that represented TMPs' meaning of caring for patients with cancer was "Restoring patients' hope in life through individualizing care." TMPs restored patients' hope through five main processes: 1) ensuring continuity in the predecessors' role; 2) having full knowledge of a patient's cancer disease; 3) restoring hope in life; 4) customizing or individualizing care, and 5) improving the patient's condition/health. Despite practice challenges, the substantive theory suggests that TMPs restore hope for patients with cancer in a culturally sensitive manner, which may partly explain why patients with cancer continue to seek their services. The findings of this study may guide research, education, and public health policy to advance traditional medicine in sub-Saharan Africa.

## Introduction

Traditional medicine practitioners (TMPs) are a critical component of healthcare systems in many sub-Saharan African countries and play vital roles in caring for patients with cancer.

**Funding:** This work was supported by The Africa Centre of Excellence for Pharm-Biotechnology and Traditional Medicine (PHARMBIOTRAC) under Mbarara University of Science and Technology, Mbarara, Uganda to AJB. The funder had no role in the study design, data collection, analysis, decision to publish, or preparation of the manuscript. No relevant grant or award recipients are specifically associated with the funding received for this study.

**Competing interests:** The authors have declared that no competing interests exist.

Limited access to medical care in poor rural or urban community settings means TMPs are primary healthcare providers for patients with cancer [1, 2]. TMPs are easily accessible for patients with cancer and are often the first to be consulted and the last provider of palliative care following discharge by orthodox doctors [1, 3–6]. Research has demonstrated TMPs offer culturally sensitive, client-centered, and holistic care for patients with cancer [1, 2]. Kayode and Victor [2] described TMPs' role in cancer care using five categories: medicine healer, spiritual guide, emotional comforter, health educator, and palliative care provider. Spiritual and psychological care provided by TMPs relieved patients' stress, anxiety, and depression [2].

Cancer care by TMPs across sub-Saharan African countries has previously been described in abstract terms. As many diseases (including cancer) in Africa are viewed as having organic and spiritual causes, most traditional care centers on managing physical and spiritual imbalances [7, 8]. In South Africa and Zimbabwe, TMPs' care for patients with cancer involved herbal medicines, spiritual care (e.g., rituals), and psychotherapy [2, 9, 10]. Healers also referred patients with cancer to other TMPs or orthodox doctors to obtain second opinions or avoid legal repercussions if the patient died.

Traditional cancer care differs across cultures. Western medicine views cancer as a somatic illness and prioritizes the reductionist-pharmacological approach, whereas traditional models emphasize body-mind-spiritual communication [7, 11]. Cancer is seen as a systemic illness caused by an imbalance in the body-mind-spiritual network [12]. Therefore, TMPs' role in diagnosing, rebalancing, strengthening, and restoring the normal functioning of that network is believed to resolve cancer. For example, the Chinese Yin and Yang philosophy is used in conjunction with traditional Chinese medicine (TCM) as a framework to manage cancer-related symptoms [12]. TCM attributes cancer symptoms to an imbalance between the Yin (calm) and Yang (sound), and TMPs restore balance using herbal medicine. Cancer is also linked to the rising qi (Sheng qi) or liver fire (anger), which animates the body and flows through the meridian system [11, 12]. Therefore, therapies such as acupuncture aim to correct this disturbance in information flow in patients with cancer [12].

Ayurveda healers in Asia attribute cancer to a loss of coordination in the three body elements (water, fire, and air). TMPs manage cancer by balancing body elements associated with specific bio-energies (e.g., Pitta: fire and water; Vatta: ether and air), improving Agni (fire), and strengthening body tissues using herbal medicine and other interventions [7, 13, 14]. Berger-González et al. [15] described cancer care provided by Mayan healers in Guatemala using body-mind-spiritual categories including herbal therapies, spiritual therapies (e.g., spiritual surgeries), physical therapies (e.g., lifestyle changes), and psychotherapy (e.g., counseling). Some traditional systems view cancer using both traditional and biomedical terms. For example, the Andean tradition categorizes illnesses, food, and medicine as hot or cold; illnesses are considered cold and treated with hot herbs, but cancer is considered a biomedical illness and managed using both modern medicine and herbs [16]. These traditional and modern explanatory models of cancer enable TMPs across various cultures to provide holistic care for patients with cancer and their families.

Despite well-developed and documented models of cancer care outside the African continent, studies conducted in Africa described TMPs' caring practices or experiences in abstract terms [2, 9, 10]. No previous studies attempted to develop a practice model to explain TMPs' perspectives of the care they provided. In addition, the caring processes and rationale behind TMPs' actions were only partially documented, making it difficult to understand TMPs' comprehensive roles in cancer care. Additionally, TMPs were reported to be unwilling to share their knowledge of cancer case management because it was based on instructions from

ancestors and varied across patients [17]. Although TMPs provide increased access to primary health care in sub–Saharan Africa and countries are increasingly recognizing and integrating them into mainstream biomedical care, the nature and quality of care provided to patients with chronic conditions such as cancer remain elusive and a public health concern [18–21]. The present study aimed to develop a substantive theory to clarify the care TMPs' provided to patients with cancer in Uganda. The findings of this study have implications for improving the quality of care in traditional medicine and its proper integration into mainstream modern medicine.

## Methods

### Study design

We used a modified Straussian grounded theory (GT) approach to develop a substantive theory around the concept of cancer care in traditional medicine as perceived by TMPs. Straussian GT has clearly laid-out procedures and techniques for inductively creating a theory [22], with coding based on the post-positivism and symbolic interactionism philosophy. Symbolic interactionism seeks to explain how people act in relation to others (interactions), as meanings (symbols) are created during these interactions [23]. Participants are believed to actively shape their social reality, rather than being passive recipients of reality. In GT research, participants create meaning through communicating with the researcher. The researcher's role is to explore, reflect, and convey participants' meanings in the emergent theory. Researchers must minimize the influence of their background, theories, knowledge, and values during data collection [22]. To ensure theoretical sensitivity (the researchers' ability to detect meaning in the data), we avoided conducting an extensive literature review before this study [22]. However, we reviewed some literature to conceptualize the phenomenon of interest and set the study scope [22].

### Setting

Data were collected from 10 districts in Western, Central, and Eastern Uganda (Mayuge, Kayunga, Wakiso, Kiboga, Gomba, Kampala, Hoima, Buikwe, Masindi, and Mukono districts) (Fig 1). These districts are inhabited by the Bantu ethnic group (Banyoro, Baganda, and Basoga tribes). Luganda is the most common local dialect spoken by these tribes and across Uganda, but other languages are used (e.g., Lusoga).

### Participants and sampling

We recruited and interviewed 18 traditional healers and herbalists with a minimum of 10 years of experience managing patients with cancer that consented to participate in this study (Table 1). TMPs had to possess relevant knowledge about cardinal signs and symptoms of at least one cancer they managed to be interviewed. Participants were herbalists (n = 12) or traditional healers (n = 6) and aged 34–64 years (n = 13) or ≥65 years (n = 5). The majority were male (n = 10), had no formal education (n = 11), and worked and lived in rural settings (n = 12). The median period of residence was 40 years (range 10–65 years). Nine TMPs were registered with their local traditional medicine umbrella organization (e.g., National Council of Traditional Healers Association; NACOTHA, https://www.nacotha.com).

Participants were initially selected using purposive sampling, with additional participants selected using theoretical sampling as data analysis progressed [24]. Theoretical sampling allows the selection of specific participants with information to develop the study concepts

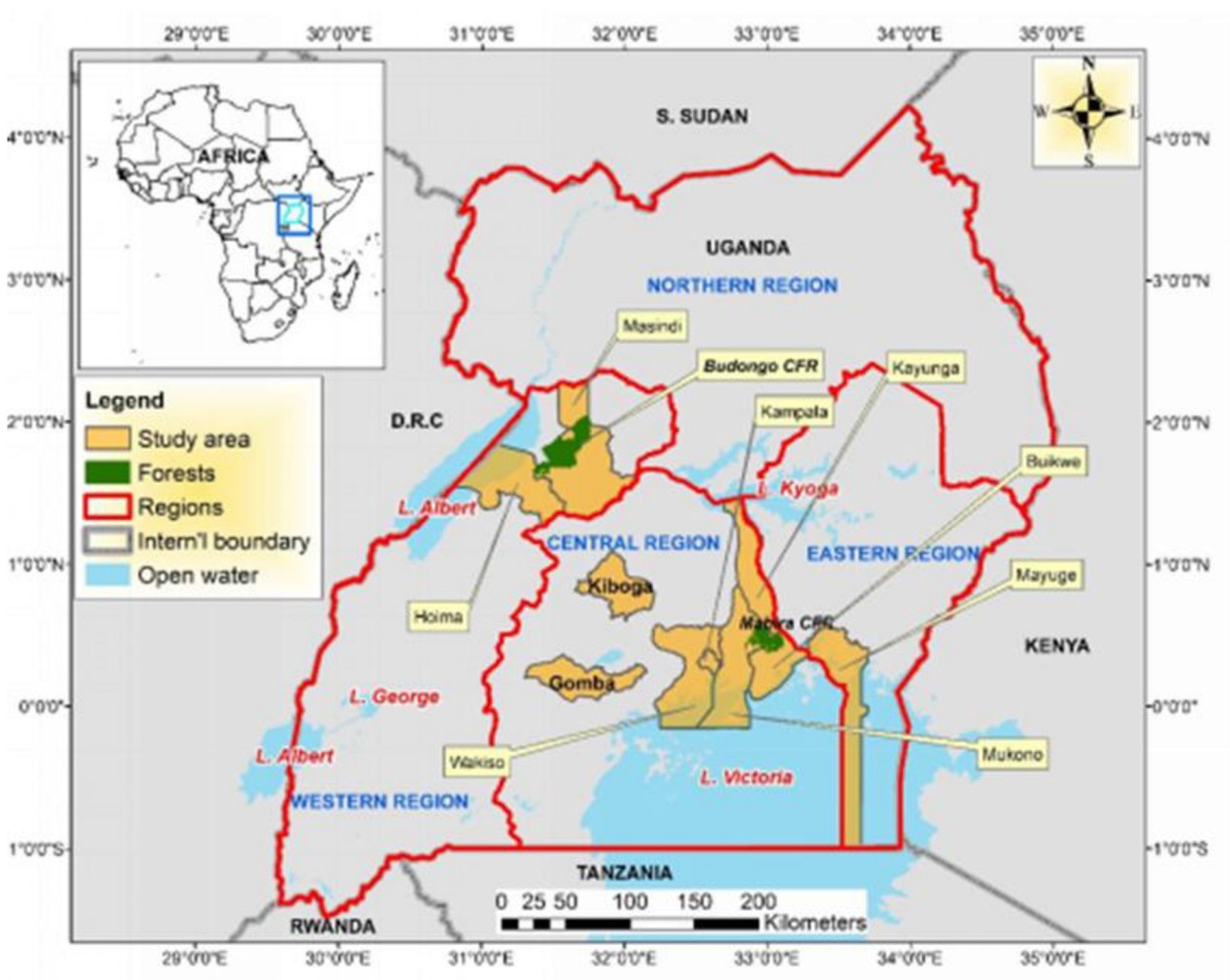

**Fig 1. Map of Uganda showing the study sites.** (ArcGIS version 10.8: 20/08/2022).

[24]. We selected further participants based on whether they were herbalists or traditional healers and whether they lived and worked in urban or rural settings until the properties and dimensions of all concepts were fully developed (theoretical saturation) [22, 25].

## Data collection

We conducted in-depth face-to-face interviews to capture TMPs' perspectives of care for patients with cancer. The semi-structured interview guide comprised four general questions that were partly based on a previous study [26]: 1) "Tell me how you know or come to know that somebody has cancer or how your clients with cancer present?"; 2) "Tell me what you do each time a patient with cancer comes to you for care in this setting?"; 3) "How do you feel about what you do in daily care for patients with cancer?"; and 4) "Describe an actual caring experience you have had with a patient with cancer in this setting that will help me understand what caring means to you?" Before data collection, these guiding questions were reviewed by an expert in qualitative research (GN) and pilot tested with two TMPs from Kampala,

**Table 1. Participants' profile.**

| Pseudonym | District | Region | Gender | Age, years | Type of TMP | Experience, years | Education | Years of residence |
|---|---|---|---|---|---|---|---|---|
| Jovia | Kampala | Central | Female | 52 | Herbalist | 40 | None | 52 |
| Sarafinah | Kampala | Central | Female | 60 | Traditional Healer | 36 | None | 60 |
| James | Kampala | Central | Male | 40 | Herbalist | 15 | Primary | 40 |
| Hajji Hassani | Mayuge | Eastern | Male | 72 | Traditional Healer | 16 | None | 20 |
| Annah | Mayuge | Eastern | Female | 45 | Herbalist | 25 | None | 45 |
| Jesica | Gomba | Central | Female | 49 | Herbalist | 30 | None | 10 |
| Karoli | Kampala | Central | Male | 75 | Herbalist | 25 | Primary | 40 |
| David | Kiboga | Central | Male | 65 | Herbalist | 10 | Primary | 65 |
| Rose | Hoima | Western | Female | 40 | Herbalist | 20 | None | 20 |
| Faridah | Kampala | Central | Female | 38 | Herbalist | 10 | Primary | 5 |
| Tereza | Masindi | Western | Male | 84 | Herbalist | 41 | Secondary | 51 |
| Silvia | Kayunga | Central | Female | 51 | Traditional Healer | 14 | None | 25 |
| Solomon | Kayunga | Central | Male | 83 | Herbalist | 50 | None | 40 |
| Jonan | Wakiso | Central | Male | 45 | Herbalist | 25 | Primary | 50 |
| Yozefinah | Gomba | Central | Female | 49 | Traditional Healer | 30 | None | 10 |
| Matia | Mayuge | Eastern | Male | 36 | Traditional Healer | 10 | None | 36 |
| Nehemiah | Buikwe | Central | Male | 39 | Herbalist | 15 | Primary | 39 |
| Stefano | Masindi | Western | Male | 50 | Herbalist | 35 | None | 50 |

Uganda's capital city. The initial interviews showed TMPs' care for patients with cancer had evolved around the dispensing of traditional medicine. Therefore, additional questions were crafted to obtain detail of TMPs' daily care for patients with cancer. For example, follow-up questions were asked about the modes of confirming a cancer diagnosis (e.g., records or medical history taking), approaches to treating and caring (e.g., diet and herbal medicine), learning how to treat cancer (e.g., from ancestors or through an experience of suffering cancer), and care for patients in special circumstances (e.g., advanced cancer, comorbidities). We added questions to differentiate cancer care from care for other nonspecific illnesses participants mentioned during their interviews (e.g., those that presented with chronic sores such as a diabetic foot) in an emergent fashion.

If participants mentioned an action or story related to caring for patients with cancer, probing questions were used to establish when, why, and how care was provided, and clarify the consequences [25]. The underlying processes and circumstances were also probed in detail [25]. To gain a deeper understanding of the care provided, hypothetical questions regarding certain caring events (e.g., managing a patient who had failed to improve) were used to stimulate discussions, especially for TMPs who found it difficult to openly discuss such events. In some cases, discussion about the care for patients with cancer continued after the formal interviews.

During data analysis, we frequently returned to previous participants to clarify the meaning of certain data that was not captured during the interviews. Interviews were audio-taped and lasted 90–200 minutes. After the development and discussion of the initial emergent theory, we returned to the original interviewees (TMPs) to confirm or validate the theory. Three focus group discussions (FGDs) were held in three districts (Kampala, Mayuge, and Masindi) comprising nine of the interviewed TMPs. The FGDs were conducted at the home of a participating TMP. Data collection and analysis were concurrent (Box 1).

Box 1. Cyclic process of data collection and analysis.

### Step 1: Open coding

- Line-by-line reading of each participant's interview transcript

- Collapsing qualitative data into their smallest components or codes

- Staying close to participants' meanings and terms and the research questions when constructing codes

- Giving names to codes using gerunds

- Organizing similar codes in clusters

- Giving names to clusters

- Rereading interview transcripts and revising initial codes

- Discussions with colleagues and coauthors

- Data collection from TMPs

### Step 2: Axial coding

- Comparing codes from interviews across different participants

- Merging codes and interview data into a single table with similar data/codes placed together

- Categorizing similar concepts or codes (forming major or elemental categories)

- Using a conditional matrix guide, process and context built into the analyses, and the relationships within each category and its codes were established.

- Revising major categories and open codes

- Data collection from TMPs

- Discussions with colleagues and coauthors

- Further abstraction of major categories to create themes

- Initial diagram and memo constructed showing the relationships and directions of the major categories

### Step 3: Selective coding

- Reflective coding matrix is used to develop the relationships and interactions among the major categories and their consequences

- Core category identified and developed

- Descriptors of the core category identified, developed, and organized

- Final flow diagram drawn

- Storyline written

- Discussions with colleagues and coauthors

- Focus group discussions with TMPs to validate the model

- Revisions made to the theoretical codes and storyline

- Substantive theory grounded in participants' data developed

## Data analysis

Interview data and field notes were transcribed in the local language (Luganda), translated into English, and then back-translated to Luganda. During transcription, initial codes and memos that originated from the data were noted. Time codes were inserted in the transcripts to enable vital parts of the conversation to be revisited or where clarity was needed during additional interviews. Data (original data transcripts, codes, and memos or notes) were entered into a password-protected Microsoft Word 2010 document. Following completion of each interview transcript, memos were written to document the participant's meaning and concepts gleaned from the data. Before analysis, transcribed interview data were read several times allowing deep immersion and conceptualization of the data. Data were then analyzed using constant comparative analysis, which involved comparing all forms of data for similarities and differences to identify boundaries, connections, or patterns [27, 28]. This cyclic approach occurred concurrently with data collection and theoretical sampling. The interviews were read meanings coded, codes compared, and gaps filled through theoretical sampling and data collection. Two constant comparison approaches were used [28]: comparing data within a single interview transcript (open coding) and comparing interviews across groups of TMPs after analyzing interviews separately (axial coding).

**Open coding.**   This involved line-by-line reading and collapsing data into small components or codes to form a basis for deeper analysis [29]. We stepped away from the data, examined segments of the interviews, and asked questions to clarify what was happening in the data or what participants' views suggested, how they came about, meaning gleaned from the data, and whose thoughts the data represented [30]. We also compared incidents with incidents or codes with codes to find differences and similarities. We focused on identifying TMPs' actions that were needed to care for patients with cancer, how such actions were sustained, and their outcomes [26, 30]. Concepts were then identified, labeled, and used to develop deeper questions, select further participants, and collect data until theoretical data saturation was reached. During open coding, we adhered to the research question as much as possible and used participants' words and gerunds when conceptualizing the data [26, 30]. (S1 Table). For example, Faridah stated:

> When you manage a cancer patient, you should be **ready to help people**, but not feel bad about them nor look down on them or **show feelings of disgust.** It is **like a calling you must be ready to serve the patient**... **[Faridah, 38 years, herbalist, Central Uganda]**

The invivo or basic codes delineated from those sentences included *Willing to help*, *Not feeling disgusted, and A calling to serve.* The basic codes were revised several times as we moved back and forth across open, axial, and selective coding. These basic codes were later

categorized under the major category- *A Calling to serve humanity* (S1 Table). In total 27 major categories were obtained from the raw data (S1 Table).

**Axial coding.** During axial coding, similar concepts or codes were categorized, linked, and organized by relationships. Relationships among the 27 major categories were established using investigative questions, reflecting on the coding process, and using a conditional matrix guide (CMG) [29, 31]. Answering the question "what category" helped to define the category using either a single participant or collective definition [31]. When populating the CMG, we used "during," "in," "because," and "by" to answer questions of when, where, why, and how the category occurred (S2 Table). The consequences of each category or event were also noted based on participants' perceptions. For example, we began with the first category, *prioritizing life over money*, and answered the question, *what is the category (prioritizing life over money)?*". Collectively, this was defined as perceiving that life is more important than money; therefore, the TMPs treated all patients regardless of their financial status. Next, we answered the question, "*when does prioritizing life over money occur?*" We found out that it occurred during general patient care and encounters with poor patients. This question was followed by, "*when does prioritizing life over money occur?*" Participants noted that this process occurred in either the TMP's clinic or the patient's home during care or management of the patient with cancer. The next question entailed the question, "*why did prioritizing life over money occur?*". Participants mentioned that the practice was an effective marketing tool because it attracted more patients with cancer through patient-to-patient referrals and that the same patients were expected to return with other illnesses. *"How did prioritizing life over money occur?"* was the fifth question answered. Participants indicated that the process occurred by providing free care, allowing patients to pay in phases, and charging less for treatment. The last question that was answered was, "*With what consequence was prioritizing life over money understood?*" Participants thought that prioritizing life over money helped them to create lasting social and economic relationships or connections. This axial coding process was repeated for all the other 26 categories (S2 Table) which enabled the revision and re-assembly of concepts and categories that were labeled and sorted under open coding to develop the structure of the caring process and an initial storyline.

The CMG was revised several times, discussed with colleagues, and compared with raw data before a memo and initial diagram were drawn (Fig 2). This diagram showed the relationships and directions of the 27 major categories that could be classified under "antecedents," "action interaction," and "consequences." Additional subthemes were created to further abstract the categories and provide a more meaningful diagram that conceptualized the initial storyline and guided what the final storyline could suggest. However, there was no core category, so selective coding was conducted after discussions with colleagues.

**Selective coding.** Selective coding involved using a core category to group categories into an "explanatory whole" that accounted for variations within categories [25]. The core category was developed using a reflective coding matrix (RCM) constructed from the major categories and their consequences in the CMG (Howell, 2006). The RCM defines the core category as a storyline or narrative that wholly describes the data [32]. In constructing the RCM, we stepped back to obtain a holistic understanding of the data. Major categories that did not appear as consequences in the CMG were temporarily set aside, as these would become dimensions in the RCM, and the core category was established using the CMG consequence categories. A core category should be sufficiently central to encompass all other categories, appear frequently, and categories should not be forced to fit it [23]. Categories that frequently appeared under the consequences section included: r*estoring hope in life*; *creating a long-lasting socioeconomic relationship*; *maintaining the patient's comfort or convenience*; and *customizing or individualizing care*. We finalized the core category by considering these frequently appearing

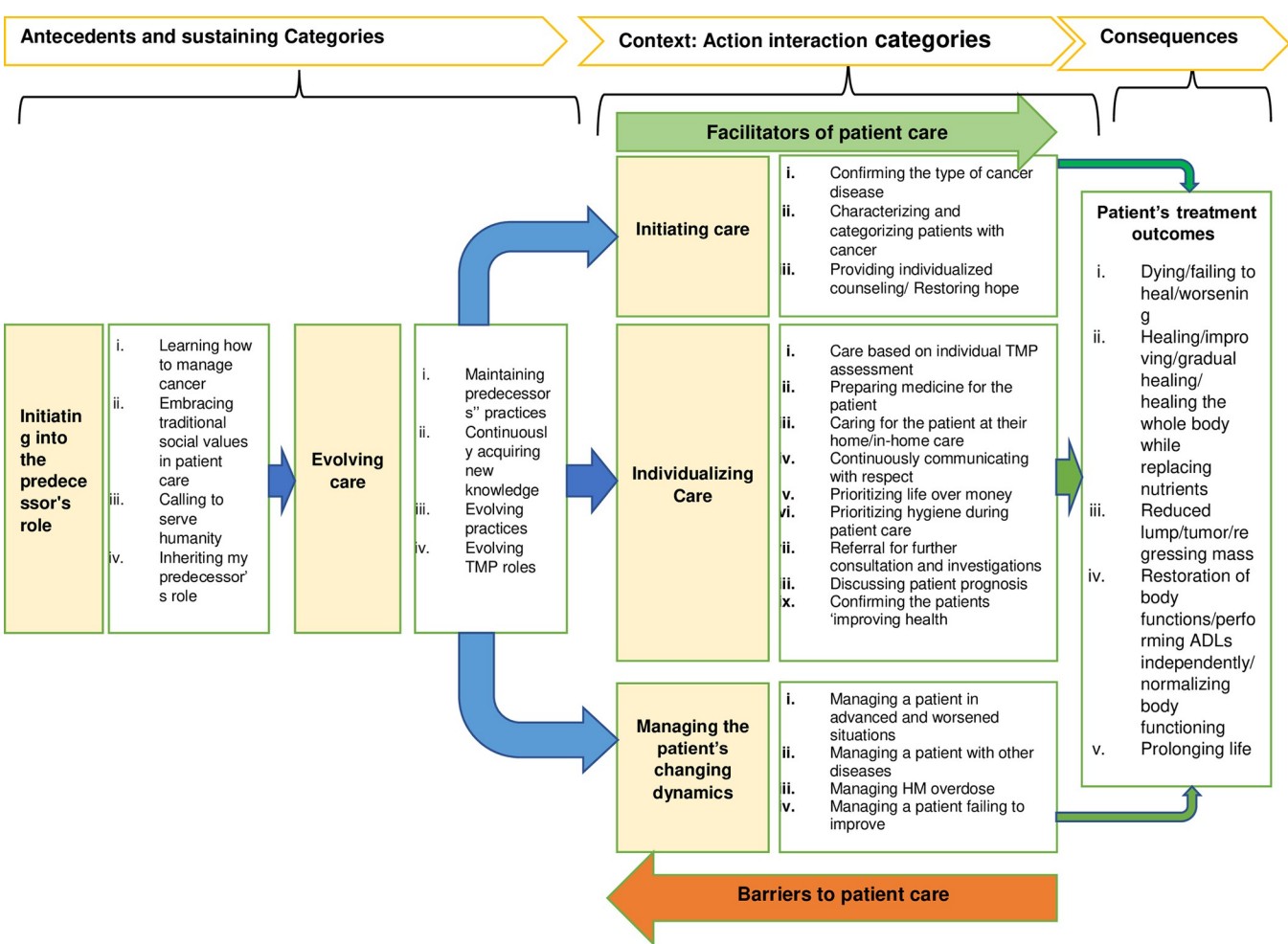

**Fig 2. Preliminary model following axial coding.** Facilitators meant all factors that enhanced care, such as the informal referral systems or use of mobile phones (technology). Whereas barriers to care included all factors that impeded care, such as the high cost of obtaining medicinal plants.

categories and answering the question, *"what is the TMP trying to do with patients with cancer?"* The working or temporary core category, *"providing hope to patients with cancer,"* appeared most frequently and aligned well with the other categories or data. To confirm the core category, we examined relevant participant quotes for similarities and differences and answered the question, *"without the core category, can caring occur?"*

We then examined categories in the CMG consequences section to identify "processes." Using the core category as a guide, these processes were used as descriptors or subcategories for the processes, properties, dimensions, context, and modes of understanding the consequences [23, 29], (S3 Table). This was accomplished by answering the question, "which of the following processes can be described as processes (actions/interactions), properties (characteristics of the category), dimensions (location of the property on a continuum), contexts, and modes of understanding of the consequences (outcomes of the process)?" Major categories that did not appear as consequences in the CMG and supporting open codes were identified and added as dimensions in the RCM.

The working core category (*"providing hope to the patient with cancer"*) could not be accomplished without *"individualized caring."* Therefore, this phrase was inserted into the

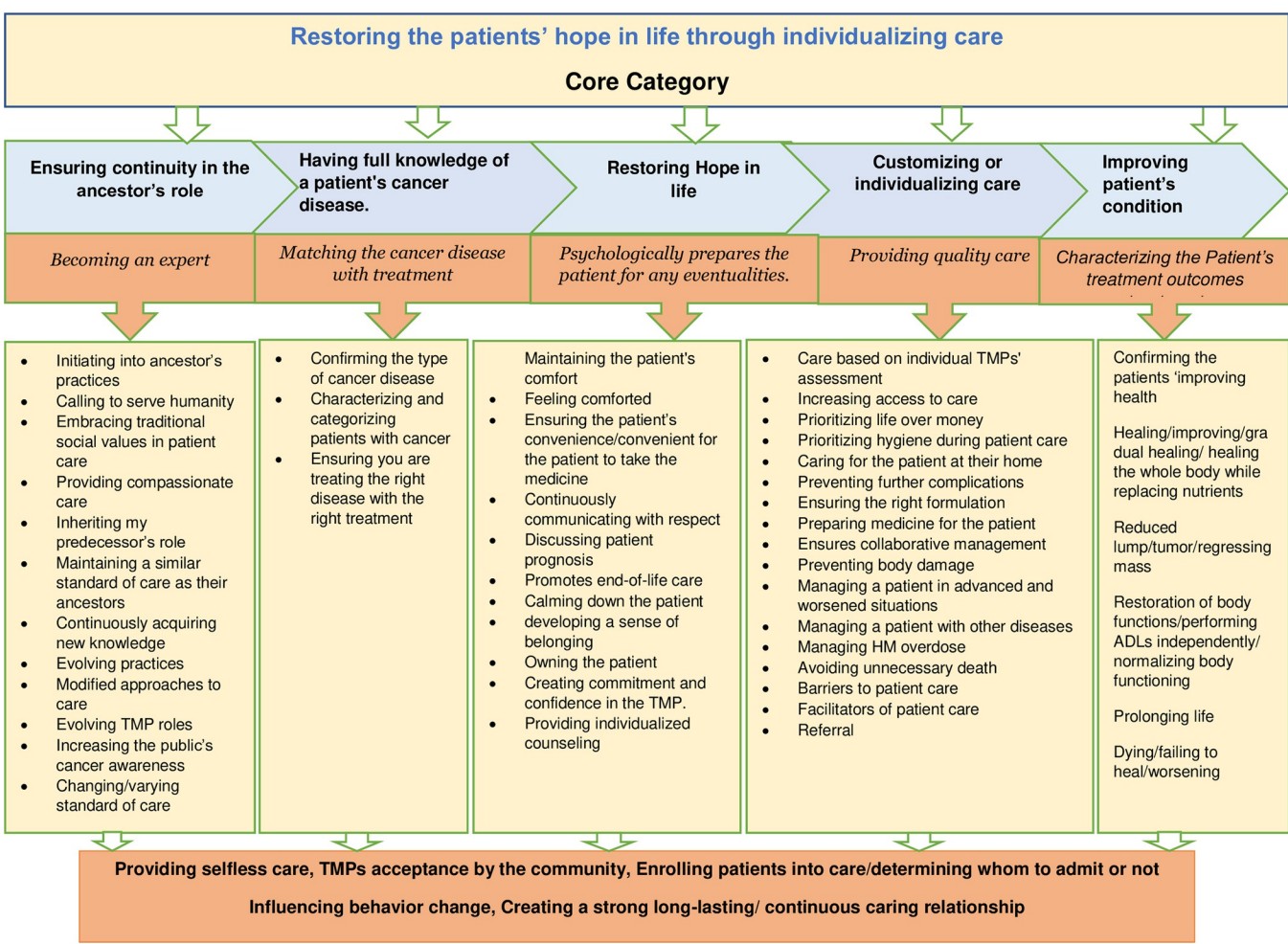

**Fig 3. Final Model conceptualizing the cancer care provided by TMPs.**

core category, which became "*providing hope through individualizing care*" (S3 Table). After completing the RCM table, we examined the relationships between the columns and read-justed their arrangement and some categories to make a meaningful flow or storyline. The storyline was concrete and able to be interpreted (from left to right) by understanding how each process supported or led to another process (S3 Table).

Continuous reflection on the RCM led to the final model (Fig 3) and a memo describing the storyline. These were reconciled by revisiting field notes, interview data, and memos and validating them with the literature and study participants (FGDs). The FGDs helped confirm the study findings and gain further insights into the preliminary theoretical categories to establish if any properties or dimensions had been missed. This provided clarity in re-wording theoretical categories and lifting some codes into categories and vice-versa.

## Ethical considerations

We conducted this study following the 1964 Declaration of Helsinki and its later amendments. The Faculty of Medicine Research Ethics Committee (FREC# 22/01-2021), Mbarara University of Science and Technology Research Ethics Board (MUREC/7#05/02-21), and the Uganda National Council of Science and Technology (Ref: HS1602ES) approved this study. Clearance

to collect data was also obtained from NACOTHA. All TMPs gave verbal and written consent to participate in this study. To minimize cross-cultural influences, FGDs were conducted separately by gender (2 = male: 1 = female). Additional information regarding the ethical, cultural, and scientific considerations specific to inclusivity in global research is included in the Supporting Information (S1 Checklist).

### Rigor and trustworthiness

Trustworthiness reflects the conceptual soundness of the research and is ensured through credibility, transferability, dependability, and confirmability [24]. Credibility or how well the collected data accurately reflected the multiple realities of the phenomenon (authenticity of the data) [24] was ensured through prolonged engagement with participants in the field (3–6 months) and data collection at multiple time points (time triangulation). Verbatim extracts from individual interviews and drafts of emergent codes, concepts, categories, and subcategories are also provided in this report. Transferability is the applicability of the findings to another setting [24]. To support transferability, we provided a full description of the research methods, interpreted results, and emerging theory. Dependability measures the stability of the data over time and under various contexts [24]. In this study, an external reviewer (GT and qualitative study expert) reexamined and verified the data (e.g., coding, emerging theory, and associated categories) to ensure GT procedures were followed. Finally, confirmability which examines the "objectivity" or neutrality of the research, was ensured through keeping an audit trail that included the raw data, codes, and memos during data collection and analysis. In addition, threats to theoretical sensitivity were bracketed through the researchers documenting (in memos) their views, beliefs, and values related to cancer and its care before and during the study to avoid imposing their understanding on study participants and their data. This allowed participants' language, meaning, concepts, and descriptions to emerge naturally [24].

## Findings

### Storyline

The storyline was constructed from the RCM and guided by four principles [32]: putting theory or abstraction at the forefront; building variation into the theory; limiting gaps; and ensuring the theory is grounded in the data. We compared the storyline with all memos and data to ensure it was a correct representation. The storyline is presented below starting with the core category, followed by subcategories. Concepts, quotes, and pseudonyms are italicized.

**Core category: Restoring patients' hope in life through individualizing care.** The core category, "restoring patients' hope in life through individualizing care," encompassed all other codes and categories. As most patients with cancer managed by TMPs had late-stage cancer, had been discharged from the hospital, and felt hopeless, TMPs' primary goal during patient encounters was restoring patients' hope. TMPs used five major processes to achieve this: ensuring continuity in the predecessors' role; having full knowledge of a patient's cancer disease; restoring hope in life, customizing or individualizing care, and improving the patient's condition/health.

TMPs indicated they were initiated into their ancestors' roles in various ways, whereby they learned how to manage cancer, obtained values related to patient care, inherited their ancestors' roles, and practiced and continued to evolve those roles. When they encountered a patient with cancer, TMPs needed to have full knowledge of that patient's cancer before they could take steps to provide hope and individualize care. Individualizing care improved the patient's condition or health, which helped to create strong, long-lasting/continuous caring relationships between TMPs and their patients, families, and friends.

**Ensuring continuity in their predecessors' role.** *Restoring patients' hope in life through individualizing care* began with *a continuation of the TMP in their predecessors'* role. TMPs perceived cancer as a complicated illness and had to *become an expert* in cancer care before embarking on their career, often on behalf of their communities. TMPs were *initiated in their predecessors' practices or roles* through: observing their predecessors treat similar patients; seeking information from predecessors; being involved in care; learning about medicine, preparation, diagnosis, and cancer severity; and experiencing cancer or being treated for cancer by their predecessors.

> *So, as children, our grandparents were so keen on us learning the medicine (or the care) knowing that once we grow up, we would take up their roles. So, he/she would say go and pick such and such medicine we boil it and give it to that person.* **[Hajji Hassani, 72 years, traditional healer, Eastern Uganda]**

> *In 1969, they referred me to a hospital to chop off my breast because it had cancer. . .However, my grandmother gave me herbal medicines while in the hospital and I healed. I also learned the medicine and I now use it.* **[Tereza, 84 years, herbalist, Western Uganda]**

In developing their caring philosophy, TMPs *embraced traditional social value*s in patient care. They treated patients with cancer with respect as if they were a member of their community or family and learned from their predecessors to treat patients with *courtesy*, *befriend the patient and show them love*, *accept and accommodate them as patients*, and *be empathetic*.

> *When patients come and explain to me their illness (cancer), I make them my friends. If they are older than me, I accept and treat them as my parents. . .* **[Sarafinah, 60 years, traditional healer, Central Uganda]**

TMPs perceived treating patients as a *calling to serve humanity*, which required feeling positive toward all patients regardless of their condition or presentation. Fear of a rebuke from God or their predecessor for obtaining the gift of treating patients for free meant that TMPs espoused a strong, warm, tolerant, and resilient spirit and positive attitude during patient care. They learned to have mercy for the sick and treat them with compassion, show no feelings of disgust, and be willing to help any patient regardless of their social, financial, and health status.

> *When you manage a cancer patient, you should be ready to help them, but not feel bad about them nor look down on them or show a feeling of disgust. It is like a calling you must be ready to serve the patient. . .* **[Faridah, 38 years, herbalist, Central Uganda]**

TMPs *inherited their predecessors' roles* by creating *a feeling of confidence and being comfortable in patient care*. This enabled them to *provide selfless care* and *attract community buy-in* during their practice years. Some TMPs *maintained their predecessors' practices of preparing medicines* in treating patients with cancer. This was often because they felt they had *settled in their ancestor's role* and were *feeling comfortable*, but also because they lacked funds and knowledge/technologies to innovate or change the ways they provided care.

> *I harvest herbal medicine and manage cancer like I was taught by my grandparents* **[Hajji Hassani, 72 years, traditional healer, Eastern Uganda]**

> *The National Drug Authority usually charges between 250,000–400,000 Uganda shillings ($68–$110) to approve our medicines, which we don't have.* **[Silvia, 51 years, traditional healer, Central Uganda]**

Adaptation to the changing social-economic environment and competition meant some TMPs *continuously acquired new knowledge* through researching medicines (trial and error), reading modern textbooks about cancer, and learning from orthodox doctors.

> *Like you have come to do research, likewise. . .I say this and this plant based on previous use by grandparents, if combined with this other plant, it can cure this type of cancer and if it works out, I give it to somebody else.* **[Jonan, 45 years, herbalist, Central Uganda]**

Some TMPs *evolved their ancestral practices* and started diagnosing and managing cancer-related comorbidities (e.g., HIV), developing and packaging standard medicinal formulas and prescriptions, adopting a business model while improving customer care, and developing innovative practices in challenging situations (e.g., delivering medicine to patients during the recent COVID-19 pandemic). *TMPs' roles had to evolve* unconsciously or consciously. Their roles had also expanded; for example, providing teaching about cancer on radio and TV (mass communication) to increase awareness, or escorting patients to hospitals for referrals where they acted as patients' caretakers.

> *I am always on the radio and television teaching people the role of food in cancer. . .God hid life in natural food. Whoever gets that food has life and whoever misses it gets sick (cancer).* **[David, 65 years, herbalist, Central Uganda]**

> *Because I have been his or her doctor, he or she becomes part of me. So, I escort him or her to the orthodox doctors. I then explain to them his or her condition and take care of them there.* **[Faridah, 38 years, herbalist, Central Uganda]**

The acquisition of knowledge, skills, and philosophy from their predecessors and subsequent role evolution meant TMPs started the care process by assessing or gaining full knowledge of a patient's cancer.

**Having full knowledge of a patient's cancer disease.** As a first step in restoring patients' hope in life, TMPs gained *full knowledge of a patient's cancer disease*. This enabled them to match treatment with the cancer type, ensured appropriate treatment dosing, and helped rule out other illnesses that presented with similar symptoms to cancer. Establishing the patient's cancer often started with the TMP *confirming the type of cancer disease*. To confirm the patient's cancer, TMPs obtained brief health histories from patients, observed and examined them for signs of cancer, and confirmed the diagnosis with the patient's medical records (often following a referral from the hospital or other traditional healers). TMPs' previous experiences of suffering or caring for patients with cancer and their intuition also contributed to confirming a cancer diagnosis.

> *It would not be good to start any patient with cancer on any other management without their medical diagnostic records because I have no machine to check. . .many diseases are similar in presentation to cancer.* **[Faridah, 38 years, herbalist, Central Uganda]**

For most TMPs, those were the most important (or only) ways they used to confirm a cancer diagnosis. However, on occasions where patients (mostly self-referrals) could not afford to obtain a formal diagnosis because of the inaccessibility of cancer diagnostic services or lack of

funds or in the absence of prior medical records, one TMP basing on his knowledge and experience, observed patients for signs and symptoms of cancer and used specific plants and animal products (e.g., bile duct) to purportedly confirm cancer.

*In the beginning, if my patient chews or swallows my medicine (bile juice or certain plants), if the person has cancer, signs will appear (like throat pain for esophageal cancer). (Rose, 40-year-old herbalist, Western Uganda).*

The second step involved *characterizing and categorizing* patients with cancer, whereby patients were sorted and described according to certain characteristics (e.g., severity, referral status, spiritual problems). To admit patients with cancer and initiate treatment, TMPs considered: 1) severe (advanced/late stage) or non-severe (early or non-advanced) cancer; 2) referrals from hospitals (often advanced cancer) or self-referrals/patient-to-patient referrals; and 3) those with comorbid spiritual problems or other challenges and those with cancer alone.

*I can know and sort out one who has a spirit and one who doesn't have it (has cancer alone). So, I know one who is possessed by a spirit by how they behave. . .I can talk to that spirit about what it wants and* where *it comes from.* **[Sarafinah, 60 years, traditional healer, Central Uganda]**

This enabled TMPs to begin restoring the patient's hope.
**Restoring hope in life.**   TMPs understood that a cancer diagnosis came with sequelae often characterized by fear. Therefore, *relieving anxiety* was paramount in forming a caring strategy and restoring patients' hope in life. This involved *continuously communicating respectfully* with the patient, often during multiple *individualized counseling* sessions. TMPs understood the importance of proper and continuous communication, especially during the early days following a cancer diagnosis. They also understood the patient's financial situation.

*The most important thing is to talk nicely with your patients, so that they have nothing to hide from you.* **[Sarafinah, 60 years, traditional healer, Central Uganda]**

*I always give patients with cancer a lot of time while talking to them. Even when they come for medicine, I always want to communicate with them daily (via the phone), to find out how they are doing and if they have taken medicine or not, and why? If they stay nearby, I visit them when their phones are off. . .* **[Faridah, 38 years, herbalist, Central Uganda]**

TMPs calmed patients who were angry and had lost hope and discussed solutions in a culturally sensitive manner. They also used indirect communication, especially when they encountered patients with poor hygiene. They averted confrontation by linking the patient's situation (e.g., poor hygiene) to wellness or the effectiveness of treatment.

*You don't tell them directly, but you attach advice on the illness. . .I tell them that if they are using my medicine, they need to be clean* **[Faridah, 38 years, herbalist, Central Uganda]**

TMPs believed patients with cancer required *extensive individualized counseling*, similar to patients with other chronic illnesses.

*A patient with cancer requires a lot of counseling like one who has HIV. Because that disease scares. . .In most cases when we get cancer, we think that it is the end of life. . .So they need to*

*take their medicine but as well must move on with their life normally.* **[Sarafinah, 60 years, traditional healer, Central Uganda]**

Clients were given options to choose from based on problems identified during care. TMPs discussed issues related to patients' health, including nutrition or diet (e.g., what food and how to prepare it), sex (e.g., when to have sex in cancer involving the sex organs), rest, lifestyle, and treatment adherence to help restore patients' hope in life. During counseling sessions, TMPs *befriended the patient*, *persuaded them*, *and allowed them* to choose among many options (rather than ordering them to adopt a specific option). This created a sense of belonging and a feeling of being comforted and loved.

Restoring hope in life for patients with advanced cancer included *psychologically preparing them for any eventualities*. Following recognition that a patient was in a severe condition, TMPs often *discussed their patient's prognosis* with the patient's family and friends to *mitigate any blame* directed toward the TMP following a patient's sudden death and *promote end-of-life care*. In such circumstances, care aimed to prolong the patient's life a little.

*After the cancer had failed to heal in the hospital, they referred to me that patient when he was weak. I told them (the patient's family) honestly that for this type of patient, I may fail to cure him because the cancer is advanced, it is in stage IV. So, if I try and fail, it is not because of me. . .* **[Stefano, 50 years, herbalist, Western Uganda]**

The processes created a sense of *ownership of the patients' situation* for both the patient and the TMP along with *a feeling of satisfaction*, *commitment*, *and confidence* in the TMP, which *influenced behavior change* in the patient such as *adhering* to the *TMP's individualized care plan* or treatment.

**Customizing or individualizing care.**   Individualized care was based on the TMP's assessment and varied from one TMP or patient to another. This assessment helped to determine when and how to manage patients with cancer and fulfilled the goal of individualized care, which was *providing quality care* to patients. TMPs prioritized life over money; a patient's life was perceived as more important than money and they treated all patients regardless of financial status. TMPs provided free care, allowed patients to pay in stages, or charged less for treatment because they believed this would be reciprocated through attracting and encouraging patient-to-patient referrals (marketing). TMPs also expected the same patients would return with other illnesses. Overall, this practice increased access to care.

*The life of an individual is the most important thing, for money even if you get a lot of money and the drum gets filled up. You will never talk with that money nor will it respect you! It is not good to have your friend (patient) suffering because of money and yet you know the plants that can help that person. . .So, I help them for free.* **[Tereza, 84 years, herbalist, Western Uganda]**

General or routine care for patients with cancer involved herbal medicine and wound care. The TMP collected, prepared, and packed medicine for the patient (often at regular intervals), especially for patients who were weak and unable to do it themselves. TMPs were cognizant of the stigma associated with patients with cancer, and prepared medicine for some patients to protect them from stigma.

*In most cases, patients with cancer are stigmatized or rejected, and people feel like those patients should not come nearer to them, especially patients with cancer of the private parts.*

*So, some patients say, "if I take many types of medicine, they will start belittling me about my illness whenever I am preparing my medicine." So, I decide as a TMP that let me mix it, so that the patient takes one jerrycan of medicine instead of many.* **[Nehemiah, 39 years, herbalist, Central Uganda]**

TMPs used *prioritizing hygiene* to prevent cross-infections when caring for patients with visible wounds, especially during/after wound care or when preparing herbal medicines. Herbal medicines were dried on tumplines as TMPs thought this *increased patients' confidence in them*. Because the cancer was believed to be contagious, TMPs wore gloves or polythene bags during wound care to avoid contracting the disease, which they later discarded. They also sun-dried the clothing they wore during wound care and avoided treating cancerous tumors during menstrual periods.

*I make sure I have gloves (during wound care) and I make sure I am clean. . .when you are clean, the patients always feel at home (comfortable) or take your medicine comfortably. . .-with a positive feeling or at peace with you.* **[Sarafinah, 60 years, traditional healer, central Uganda]**

Care was customized for patients with severe illness or advanced-stage cancer, including providing care at patients' homes (*in-home care* or *home-based care*). TMPs thought that such patients suffered *fatigue* upon moving to their clinic, which affected their ability to evaluate the *patient's improvement*. Therefore, if the patient or their family could afford to pay the TMP's fees (e.g., transport and upkeep), they offered care and support at the patient's home. This often involved observing or assessing the patient, providing counseling, and herbal medicine. TMPs commented that in-home care prevented further complications due to fatigue associated with continuous movements.

*Sometimes they call me on [the] phone and send me transport fees to see the patient. I go along with the medicine and monitor them, especially when they are weak. Because if they are weak and then they add on the burden of coming to see me, they get very tired, and you may not know if they are improving or not.* **[Stefano, 50 years, herbalist, Western Uganda]**

Caring for patients with cancer was often complicated by challenges. TMPs had to deal with multiple patient realities, and manage *patients in advanced and worsened situations*, *patients with other chronic diseases*, *patients who had taken an overdose of herbal medicine*, and *those they treated but had failed to improve*. TMPs often cared for patients with advanced and worsened conditions using plant remedies, including tapering, and later increasing their anticancer herbal dose. They used intuition and experience to recognize if a patient was not improving and referred that patient to orthodox doctors; this decision was often made jointly with the patient's relatives. TMPs also assessed and treated (when they could) patients with comorbid chronic illnesses or referred them to orthodox doctors to enable fast healing of cancer.

*You put him on herbal medicine for cancer and then she gets drugs for hypertension elsewhere (hospital).* **[Stefano, 50 years, herbalist, Western Uganda]**

Sometimes TMPs treated patients who had taken an *overdose of herbal medicine*. In such cases, TMPs took steps to prevent the patient from worsening, including establishing how the herbal medicine was taken, teaching patients about the correct use of medicines, and ordering patients to stop and then later restart treatment after receiving proper instructions.

*I gave herbal medicine to one patient who almost died of it. You know patients have a belief that African herbal medicine has no dose, so she took it and almost died. I told her to take a lot of water and stop taking medicine for about 3 days and continue later to take it as pre-scribed.* **[Karoli, 75 years, herbalist, Central Uganda]**

Furthermore, based on their experience, on observing that their patients were not improv-ing, TMPs investigated and discussed reasons for the failure to improve, guided the patient about possible solutions, and referred them to other TMPs or orthodox doctors as necessary. Because of a lack of technical abilities and the need to form a basis for further management, TMPs often referred patients to orthodox doctors with instructions to obtain diagnostic details or check for any signs of improvement.

In addition to various *patient dynamics*, TMPs encountered *barriers to patient care* that often limited the *patient's access to care* because of increased costs, such as increased transport costs to obtain medicine (because of deforestation), increased cost of care if a patient was "dumped" at the TMP's home, high taxes, high advertising costs, and religious views against traditional medicine.

*Most of them are poor and cannot afford to pay. You give them the medicine they cure, they go, and they don't return, they usually don't pay, and yet I have put in my money to go and fetch the medicine far away. So today, some TMPs don't bother to help. . .* **[Jonan, 45 years, herbalist, Central Uganda]**

TMPs also described *facilitators of care* that *enhanced access to care* for patients, such as pro-viding free or cheap care and allowing patients to pay in stages. They worked to create an envi-ronment that allowed the importation and delivery of medicine, telephoned patients and delivered medicine (technological advancement), treated patients irrespective of their faith, and informally collaborated with orthodox doctors. TMPs noted that because many patients they cared for had advanced cancer and had been discharged from modern treatment, individ-ualized care intended to offer hope and improve the patient's condition/health.

**Improving patients' condition/health.** Most TMPs believed individualizing care helped patients to improve or heal. To confirm *patients' improving health*, TMPs sought information about changes in the patient's condition following treatment initiation. Confirming patients' improving health signified the treatment's effectiveness, assisted in further decision-making, and created a *positive feeling of satisfaction* for the TMP. Improvement was often determined through asking questions, the patient describing signs of improving health, examining the patient, and referring patients to the hospital to confirm improvement using orthodox doctors.

*The patient who improves will tell you, but we also examine them to see if they have improved.* **[Yozefinah, 49 years, traditional healer, Central Uganda]**

*If a patient came when they are swollen and now, they are not and they have not died. If they were not walking and now, they walk, what other evidence of healing will you need apart from that? But for me to confirm that they have healed they have to go back to the hospital and investigations are done to prove that they have healed.* **[Silvia, 51 years, traditional healer, Central Uganda]**

Following confirmation of the change in the patient's condition, TMPs *characterized their patient's treatment outcomes*, which could aid *in determining prognosis in future patients* or

determine whom to admit in the future. TMPs assessed, described, and gave reasons for various possible outcomes. However, each TMP described their outcomes in different terms that were often contradictory. Most TMPs mentioned the dimensions of improving health as ranging from *dying/failing to heal/worsening* to *healing/improving/gradual healing/healing the whole body while replenishing the body's nutrients*. Patients with early-stage cancer and those that adhered to the TMP's individualized care plan were more likely to have better outcomes than those who had advanced cancer or did not adhere to their treatment plan.

*If we find advanced cancer, there is a likelihood of dying from cancer despite using the same treatment. But if the person has early-stage cancer the person is likely to cure.* **[Silvia, 51 years, traditional healer, Central Uganda]**

Based on the TMPs treatment experience and observation, other common outcomes included reducing or regressing cancerous wounds or masses, prolonging life, restoring body functions, or independently performing activities of daily living, and normalizing body functioning.

*Our treatment aims to heal somebody and their whole body so that it returns to normal functioning, as God created it.* **[Jonan, 45 years, herbalist, Central Uganda]**

Overall, TMPs provided individualized care as a means to restore hope, which they believed helped patients improve. Improvement ensured strong long-lasting/continuous caring relationships that lasted beyond the patient's current illness and extended to other patients through patient-to-patient referrals. In addition, it allowed the patient to continuously interface with the TMP when they faced other health challenges.

## Discussion

This study investigated TMPs' meaning of caring for patients with cancer in Uganda and proposed a practice model to develop an understanding of traditional care processes for patients with chronic illnesses such as cancer. The substantive theory of *restoring the patients' hope in life through individualizing care* offers a new lens for visualizing care by TMPs in sub-Saharan Africa, with multiple learning points and policy implications for mainstream modern and conventional healthcare systems. The study findings are largely discussed using extant and collateral literature from indigenous and conventional medicine around our major categories or concepts to validate or refute our model, because of limited literature around similar practice models in sub-Saharan Africa.

### Ensuring continuity in the predecessors' role

We found TMPs had learned how to manage cancer during their pre-practice years as it was perceived as a complicated illness. This finding was consistent with an earlier study conducted in Tanzania [33]. Previous studies from Zimbabwe and South Africa showed TMPs generally assumed the healing role before receiving any basic training, often through personal choice or a calling via dreams or spiritual possession (or sickness) by God or the spirit of a deceased predecessor [34, 35]. Apprentices underwent training with a master healer, who was responsible for instructing the apprentice in both sciences (e.g., physical care) and mysticism (e.g., spiritual care). The apprentice was then able to treat illnesses (cancer inclusive) based on mystical and scientific knowledge and manage patients' body-mind-spiritual imbalances [34, 35].

Participating TMPs indicated that having skills alone was not sufficient, as they had to be rooted in traditional social values and accept their role as a calling to serve humanity. Most social values embraced by TMPs in this study originated from "Ubuntu philosophy," which is an ancient African worldview of rational ethics based on community values (e.g., compassion, humanness, caring, respect, dignity, and solidarity with others) [36]. Ubuntu defines patients in terms of the community and required respect and care; therefore, TMPs had a conscious or unconscious obligation to provide selfless care, which was a minimum expectation of patients with cancer [36, 37].

Leininger's transcultural theory suggests harmful cultural values should be discarded or discouraged, beneficial values embraced, and values negotiated when they are neither harmful nor beneficial [38]. Our finding related to cultural values, therefore, has educational and health policy implications for sub-Saharan African countries. Currently, the training of general medical practitioners in Uganda still largely emphasizes the medical model and not holistic and culturally sensitive care (bio-psycho-social care) which are relevant to the modern oncological approach to the management of patients with cancer [39, 40]. This may partly explain why patients with cancer prefer to use the services of TMPs or even abandon medical care for traditional medicine leading to late referrals to hospitals and poor prognosis [41]. Importantly, recent anecdotal evidence suggested traditional social values may be on the verge of extinction because of an increasingly individualistic society [42]. Our findings can therefore guide the training of new traditional and Western medicine healthcare providers, including pharmacists, pharmacognists, nurses, and doctors working with patients with chronic diseases such as cancer. Traditional social values explained by TMPs in this study can be integrated into education curricula and training for future healthcare practitioners in Africa. This will support the provision of culturally sensitive care, which is often lacking because education systems are Westernized and emphasize Western values above African social values [43, 44]. All healthcare workers need to embrace and observe traditional social values such as friendliness, humaneness, compassion, and solidarity with the patient. These African meanings of the Ubuntu values may need to be added during healthcare workers' initial or refresher training to help bridge this gap and improve the quality of patient care.

Our findings also highlighted that learning to manage patients with cancer evolved throughout TMPs' practice years, which enabled them to practice their predecessors' roles differently. This was consistent with a study from Nigeria that found TMPs used new methods to sustain old knowledge and ideas [34]. The evolution of TMPs' roles, as shown in this study, may require minimum standards of traditional practice to be established to support the growth of traditional medicine in African settings.

## Having full knowledge of a patient's cancer disease

Cancer often presents with nonspecific symptoms, so TMPs confirmed the diagnosis before starting to restore hope for their patients, which was consistent with a report focused on general patients from Tanzania [33]. A study from Bolivia showed healers referred patients with cancer to modern doctors and used their diagnosis as a basis to establish the illness severity [16].

Importantly, TMPs in this study used animal or plant products to diagnose cancer for some patients who had less access to modern diagnostic services. Similarly, a Tanzanian study showed additional methods of diagnosis were divination or learning about the patient's disease from ancestors through rituals and dreams [29]. Unlike orthodox doctors, this improvisation by TMPs and the ability to explain and reassure the patient psychologically about their diagnosis (cancer disease) in the face of scarcity of modern diagnostic services may explain why patients with cancer may trust healers more than orthodox doctors.

## Restoring hope in life

Participating TMPs understood that a cancer diagnosis came with sequelae characterized by fear; therefore, relieving anxiety was important for a caring and hope restoration strategy. As beliefs influence the body through mind-body interactions [11], TMPs in Africa were reported to offer false hope following diagnosis with chronic illnesses, such as cancer and HIV [45]. In Nigeria, direct feedback from orthodox doctors was perceived to create hopelessness, whereas TMPs maintained that chronic illnesses (e.g., cancer) were caused by spirits and could be healed [45]. This message resonated with patients and enhanced cooperation with TMPs' instructions [46]. TMPs cognitively reconstructed patients' reality by providing false hope so they could pay attention to the future, find meaning in life, and relieve depression and anxiety [47, 48]. In this study, TMPs addressed anxiety through continuous respectful communication and multiple individualized counseling sessions. This was consistent with studies conducted in Uganda, South Africa, and Malaysia, which demonstrated TMPs provided personalized counseling, including for families, unlike orthodox doctors [8, 49, 50]. In addition, TMPs understood their patients' beliefs, values, behavior, and language, meaning they counseled their patients from a cultural perspective that aided the uptake of their counseling messages [50].

Previous research in Uganda and Denmark indicated that although healthcare workers can provide hope to patients with cancer, limited time means such psychosocial support is usually inadequate [46, 47]. Communication between healthcare workers and patients with cancer in Uganda was inadequate, with many patients not receiving full information about their illnesses [40, 46]. Our emerging theory suggests oncological nurses, doctors, and TMPs in Africa need to allocate more time for patients with cancer during counseling sessions, listen and counsel patients beyond their medical issues, provide personalized care, and be culturally sensitive while restoring patients' hope.

Psychologically preparing patients with advanced cancer for any eventualities through discussing their prognosis was another way TMPs restored hope, which avoided community backlash if the patient died and allowed patients to plan for their end-of-life. Ubuntu philosophy suggests right actions promote friendly relationships with patients, families, and communities [37]. TMPs were therefore obliged to tell the truth when discussing the patient's prognosis if it promoted the survival and harmony of TMPs within their community. A Jordanian study indicated false reassurance was a cultural norm, and healthcare workers minimized bad news when sharing prognostic information through false reassurance and hope [51]. In Kenya, doctors also hid prognostic information from patients to maintain hope [52]. TCM practitioners restore patients' hope by showing compassion and positive intent to heal to induce beneficial psychological adaptation during treatment [12]. These findings highlighted that giving prognostic information in a culturally sensitive manner is an effective tool in restoring hope in life among patients with cancer.

## Customizing or individualizing care

Although TMPs' care for patients with cancer in this study varied across patients, the meaning of care was typified by prioritizing life over money. Similar studies from Uganda showed that TMPs were not motivated by money, attended to patients immediately with/without money, and allowed patients to pay later (e.g., in installments or in-kind) [53, 54]. A study from Bangladesh reported healers often demanded nothing for their services or fixed no fees and accepted whatever token of appreciation that was offered [55]. In Africa, Ubuntu philosophy emphasizes reaching out to others in need through friendship, generosity, hospitality, caring, and sharing what one owns. This means TMPs were obliged to be friendly, promote friendship, and avoid unfriendliness [36, 37]. Therefore, any acts involving rejection, abrupt

discharge, discontinuation of medicine, or providing inadequate care for patients that had little/no money (as often practiced by orthodox doctors) would be seen as unfriendly and lead to community rejection of the TMPs [54].

In this study, TMPs individualized care for patients with cancer, such as providing in-home care for those with advanced cancer. Ayurveda provides a framework to explain the individualization of patient care in traditional medicine, as each person is believed to have a unique body constitution (Prukruti) reflecting the balance of three bio-energies (Pitta, Kapha, Vata) that dictate one's predisposition to certain illnesses (including cancer) and personal qualities [11]. Therefore, treatment is tailored to an individual's unique characteristics.

Although TMPs reported various facilitators that enhanced access to individualized care, such as providing free/cheap services and using informal referral networks, they found caring for patients with cancer was sometimes challenging, especially given the increased costs of providing care (obtaining herbal medicine) because of environmental degradation and other economic factors. These challenges affected TMPs' adherence to traditional values of care. Therefore, providing capital to develop land or skills in medicinal species preservation is paramount. In addition, because TMPs informally learned from and referred patients to Western doctors, which facilitated care. This suggests it is necessary to formalize collaboration between orthodox doctors and TMPs to harness the mutual benefits.

## Improving patients' condition/health

Participating TMPs claimed that adherence to individualized care provided hope and improved the patient's condition/health. However, this improvement had to be confirmed, which enabled the characterization of the patient's treatment outcomes. Unlike in Western medicine where cancer outcomes are described in objective clinical and patient-centered terms, TMPs described various dimensions of improving health using subjective terms (e.g., healing the whole body while replenishing the body's nutrients). Similarly, a previous systematic review reported several outcomes of Chinese traditional medicine that included prolonged life, improved recovery times and symptoms, and accelerated wound healing or healing times [55]. Further research is warranted to evaluate, confirm and translate these outcomes in other informal settings and chronic illnesses treated by TMPs.

## Strengths and limitations of this study

Using GT methodology, this study proposed one of the first substantive theory around the traditional meaning of care for patients with chronic conditions (e.g., cancer) by TMPs in sub-Saharan Africa (SSA). This theory may guide future research in similar cultural settings.

However, this study had limitations that must be considered. Because culture is dynamic and unique and that this study took a solely qualitative nature, generalization of the study findings to some cultures in Africa may be challenging. In addition, we did not include TMPs with less than 10 years of experience because of the need to obtain credible information; therefore, the findings may not fully represent new entrants who may not be traditionally trained but provide care based on modern capitalistic or business values or those traditionally trained elsewhere (abroad) and may blend traditional and foreign cultural values and principles in the care of patients with cancer. Second, we did not interview patients with cancer receiving traditional care. Comparison of patients' experiences with those of the TMPs could have provided further insights and enriched our substantive theory. GT researchers validating the model in further studies could include patients in constant comparative analyses.

## Conclusions

This study aimed to explore the TMP's meaning of caring for patients with cancer and develop a substantive theory to clarify that care, using ethnographic interviews and constant comparative analyses. Generally, the substantive theory indicates that TMPs' traditional meaning of care for patients with cancer involved moving patients from a state of hopelessness to one of hope, as encapsulated by the core category, "*restoring the patients' hope in life through individualizing care*." Our findings related to the substantive theory have research and public health policy implications for sub-Saharan African countries.

Despite the continued recognition of TMPs' role in healthcare across Africa, much of the focus has been on standardizing traditional medicine rather than improving the traditional care system. Our emerging theory provides a framework to assess the quality of services provided by TMPs, especially those that have evolved their practices. Standardized tools and checklists could be developed from this model to support the improvement of traditional systems.

Further GT research focused on cancer or other chronic illnesses in other cultures in sub-Saharan Africa is warranted to develop a grand theory to explain traditional care for patients with chronic diseases in this context. In addition, using our theory as a guide, further studies in African settings could explore the healing practices and experiences of new and internationally trained healers working in modern settings and develop and compare their models of care with traditionally-trained TMPs. Future studies may also explore the subject of care in traditional medicine using the mixed methods GT which would allow concurrent generation and testing of the GT as well as linking theory to practice.

Lastly, despite practice challenges, the substantive theory suggests that TMPs restore hope for patients with cancer in a culturally sensitive manner. There is a need to formalize and harness the mutual benefits of collaboration between orthodox doctors and TMPs in sub-Saharan African settings. This will provide a platform for orthodox doctors to learn from TMPs and vice versa and improve patient outcomes.

## Supporting information

**S1 Checklist. Inclusivity in global research checklist.**
(DOCX)

**S1 Table. Open codes and supporting transcripts.**
(DOCX)

**S2 Table. Axial codes (Conditional matrix guide).**
(DOCX)

**S3 Table. Selective codes (Reflective coding matrix).**
(DOCX)

## Acknowledgments

The authors express sincere gratitude to NACOTHA and the traditional medicine practitioners who provided the data reported in this study, and Ms. Audrey Holmes who copyedited a draft of this manuscript. Special thanks also go to Dr. Grace Nakate, Mr. Moses Wachiri, and Ms. Mary Namuguzi of Aga Khan University for their insightful input and feedback on the paper.

## Author Contributions

**Conceptualization:** John Baptist Asiimwe, Prakash B. Nagendrappa, Esther C. Atukunda, Grace Nambozi, Casim Umba Tolo, Patrick E. Ogwang, Maud M. Kamatenesi.

**Formal analysis:** John Baptist Asiimwe, Grace Nambozi.

**Funding acquisition:** John Baptist Asiimwe, Prakash B. Nagendrappa, Esther C. Atukunda, Casim Umba Tolo, Patrick E. Ogwang, Maud M. Kamatenesi.

**Investigation:** John Baptist Asiimwe, Grace Nambozi, Casim Umba Tolo, Patrick E. Ogwang.

**Methodology:** John Baptist Asiimwe, Prakash B. Nagendrappa, Esther C. Atukunda, Grace Nambozi, Casim Umba Tolo, Patrick E. Ogwang, Maud M. Kamatenesi.

**Supervision:** Prakash B. Nagendrappa, Casim Umba Tolo, Patrick E. Ogwang, Maud M. Kamatenesi.

**Writing – original draft:** John Baptist Asiimwe, Prakash B. Nagendrappa, Esther C. Atukunda, Grace Nambozi, Casim Umba Tolo, Patrick E. Ogwang, Maud M. Kamatenesi.

**Writing – review & editing:** John Baptist Asiimwe, Prakash B. Nagendrappa, Esther C. Atukunda, Grace Nambozi, Casim Umba Tolo, Patrick E. Ogwang, Maud M. Kamatenesi.

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
