## [Decision Letter · Decision Letter 0]

28 Apr 2023

PGPH-D-23-00359

The Meaning of caring for patients with cancer among traditional medicine practitioners in Uganda: A grounded theory approach

Dear Dr. Asiimwe,

Thank you for submitting your manuscript to PLOS Global Public Health. After careful consideration, we feel that it has merit but does not fully meet PLOS Global Public Health’s publication criteria as it currently stands. Therefore, we invite you to submit a revised version of the manuscript that addresses the points raised during the review process.

The two reviewers have provided valuable suggestions to improve the current version of the manuscript. I would invite you to respond to these suggestions and revise your manuscript.

We look forward to receiving your revised manuscript.

Kind regards,

Mathew Sunil George

Academic Editor

Journal Requirements:

Additional Editor Comments (if provided):

Reviewers' comments:

Reviewer's Responses to Questions

**Comments to the Author**

1. Does this manuscript meet PLOS Global Public Health’s publication criteria? Is the manuscript technically sound, and do the data support the conclusions? The manuscript must describe methodologically and ethically rigorous research with conclusions that are appropriately drawn based on the data presented.

Reviewer #1: Yes

Reviewer #2: Partly

2. Has the statistical analysis been performed appropriately and rigorously?

Reviewer #1: I don't know

Reviewer #2: Yes

3. Have the authors made all data underlying the findings in their manuscript fully available (please refer to the Data Availability Statement at the start of the manuscript PDF file)?

Reviewer #1: Yes

Reviewer #2: Yes

4. Is the manuscript presented in an intelligible fashion and written in standard English?

Reviewer #1: Yes

Reviewer #2: Yes

5. Review Comments to the Author

Reviewer #1: This paper provides very interesting research work addressing the The Meaning of caring for patients with cancer among traditional medicine practitioners in Uganda: A grounded theory approach

1. The introduction requires substantial improvement.

2. Can you change the title of your paper?

3.  I would suggest to include more literature if possible.

4. Is there any other theory of the research framework used in your study?

5.  Can you include gender aspects in your study?

6. Is there any other limitations of your study?

7. What is the strength of your study?

8. How your paper is more relevant to public health aspects ?

10.Future recommendation: Can you try a mixed-method approach for the future study?

11. Discussion section needs minor improvement.

Reviewer #2: This research into the Meaning of caring for patients with cancer among traditional medicine practitioners

in Uganda: A grounded theory approach, is insightful and well grounded. It is potentially a good paper that contributes original knowledge to an otherwise under-researched area, particuloarly with regards to social scientific studies into cancer care in low-resource settings.

However, while i really enjoyed reading your paper, i feel that the article could do with improved organisation of material, tightening up and emphasis on key areas, and getting rid of repetitive material especially towards the end of the paper.

I will now detail what critical revisions are necessary to improve the overal quality of this paper:

1.ORGANISATION: the paragraph starting on Line 95 should be moved up either at the beginning of the introduction or immediately after paragraph one in the introduction. This is simply to enhance the coherence of this section. Please see more on organisation at the end of this review.

2. METHODOLOGY: While I appreciate the authors nicely explaining the GT approach and how it has been applied to data collection and analysis, this section needs further work to tighten it up and make it much more accessible to non expert readers. Isuggest eliminating the tables and either, integrating the text into the body of the work, or using simple text boxes to highlight important coding principles, while not clogging up the boxes to the extent that they are difficult to follow as is currently the case.

Secondly, I am not convinced that TMCs in this study were always able to distinquish between cancer symptoms and other illnesses because a) the authors confirm that TMCs have no formal education and b) it is not clear whether or not they only deal with medical referrals or any other suspected cases of cancer. I suggest reframing these claims to something along the lines of ' TMCs evaluated symptoms based on own knowledge and experience and had xyz criteria for arriving at a cancer diagnosis in the absence of prior medical records...' This is because in my own experience researching indigenous healing systems, TMCs always engage in trial and error practices and often rely on a repertoire of historical illness symbols to assign symptoms to non-specific category of illness. For example, when reseraching fever and febrile illness in Tanzania, i found that clusters of symptoms categorised as fever included at least five out of 8 well known categories that were passed on through generations.Additionally, treatment was not alway strictly traditional, rather, some form of hybrid between allopathic and herbal medicine, and, as you rightly point out, suggestions for diet and lifestyle changes on the part of the patient and his or her carers. I would like to see these nuanses brought out more in your analysis so that it does not look like TMC cancer care is a linear and homogenous practice across the districts. I am sure you would agree that this is not the case.

3: ANALYSIS ( MINOR CHANGES)

I suggest expanding finding number 2 to incooperate a section on what happens when patients do not have medical records confirming cancer. What would the TMC typically do? How would they arrive at a diagnosis? I am aware that in low resource settings the vast majority of cancer cases are either never diagnosed or diagnosed at very late stages and therefore often there is no cure. It will benefit the paper to talk briefly about this.

LINES 536-540 seem contradictory. You state that TMCs in your study believed that herbal remedies were superior to allopathic treatmentS, yet, they send patients they couldn't cure back to allopathic doctors? Why was that? This needs a few lines to clarify, and supported by vignettes. See similar contraditions on the page that follows this (Lines 553-557). Also please substantiate Lines 602 onwards. How did TMCs know that tumors were reducing if by own admision they said they couldnt check these with 'machines'? I suggest rewording with something less grand...

4. DISCUSSION Lines 613-723 reads like a literturte review, and is mostly repetitive of your findings. I suggest reworking this section, bearing in mind the original research questions and whether or not you feel you have answered them, and or how your proposed substantive theory has been applied to the findings, and the contribution this makes to existing literature ( weaving in the litertaure review as you sometimes do).

5. GENERAL COMMENTS:

Lines 725-727 should move to the literture review and then referered to back in the implications. Often, it is a distraction to include new literature at the end of a paper.

Lines 733-735 requires support with references, and so do lines 744 to 748.

Lines 761-766 is a key discussion point and should come in the DISCUSSION section and preceded in the litertaure review. I.e a small section detailing how TMCs training differs from orthodox doctors in Uganda , and the implication of this for managing chronic illness such as cancer in rural under-resourced settings.

In STUDY LIMITATIONS, Lines 772 -774 need a brief substantiation. Are you suggesting that newer entrants into traditional healing praxtices are going in it for money? Is this a sense you got from your research or your own views? It would be important to know what makes you feel this is the case.

Finally the conclusion should typically talk about how you have answered your research question/s, what this means for further research and how your proposed tehory can further be applied to similar studies moving forward. At present your conclusion is but a brief summary of your findings.

6. PLOS authors have the option to publish the peer review history of their article (what does this mean?). If published, this will include your full peer review and any attached files.

**Do you want your identity to be public for this peer review?** For information about this choice, including consent withdrawal, please see our Privacy Policy.

Reviewer #1: **Yes: **Saddaf Naaz Akhtar

Reviewer #2: **Yes: **Dr Violet Barasa

---

## [Decision Letter · Decision Letter 1]

15 Jun 2023

The Meaning of caring for patients with cancer among traditional medicine practitioners in Uganda: A grounded theory approach

PGPH-D-23-00359R1

Dear Dr. Asiimwe, 

We are pleased to inform you that your manuscript 'The Meaning of caring for patients with cancer among traditional medicine practitioners in Uganda: A grounded theory approach' has been provisionally accepted for publication in PLOS Global Public Health.

Best regards,

Mathew Sunil George

Academic Editor

Reviewer Comments (if any, and for reference):

Reviewer's Responses to Questions

**Comments to the Author**

1. If the authors have adequately addressed your comments raised in a previous round of review and you feel that this manuscript is now acceptable for publication, you may indicate that here to bypass the “Comments to the Author” section, enter your conflict of interest statement in the “Confidential to Editor” section, and submit your "Accept" recommendation.

Reviewer #1: All comments have been addressed

Reviewer #2: All comments have been addressed

2. Does this manuscript meet PLOS Global Public Health’s publication criteria? Is the manuscript technically sound, and do the data support the conclusions? The manuscript must describe methodologically and ethically rigorous research with conclusions that are appropriately drawn based on the data presented.

Reviewer #1: Yes

Reviewer #2: Yes

3. Has the statistical analysis been performed appropriately and rigorously?

Reviewer #1: Yes

Reviewer #2: Yes

4. Have the authors made all data underlying the findings in their manuscript fully available (please refer to the Data Availability Statement at the start of the manuscript PDF file)?

Reviewer #1: Yes

Reviewer #2: Yes

5. Is the manuscript presented in an intelligible fashion and written in standard English?

Reviewer #1: Yes

Reviewer #2: Yes

6. Review Comments to the Author

Reviewer #1: Overall, this is a clear, concise, and well-written manuscript

Reviewer #2: Dear authors,

well done on addressing the questions and suggestions raised during our review of your first submission. I suggest acknowleging the lack of gender dimensions in your research as this is quite important in social scinece research of this nature. Questions of how for example, an inclusion of gender factors would impact your intepretation of findings, how TMP approach care for female vis a vis male patients and their families etc are hugely significant as they bring out the nuances of care that are missed in generalised biomedical studies.

7. PLOS authors have the option to publish the peer review history of their article (what does this mean?). If published, this will include your full peer review and any attached files.

**Do you want your identity to be public for this peer review?** For information about this choice, including consent withdrawal, please see our Privacy Policy.

Reviewer #1: **Yes: **Saddaf Naaz Akhtar

Reviewer #2: **Yes: **Dr Violet Barasa
